# Molecular Dynamics Study of Structural and Transport Properties of Silver Iodide Using Effective Charges

**DOI:** 10.3390/molecules27186132

**Published:** 2022-09-19

**Authors:** Diego Peña Lara, Hernando Correa, Jesús Evelio Diosa

**Affiliations:** 1Grupo de Transiciones de Fases y Materiales Funcionales, Departamento de Física, Cali 760 032, Colombia; 2Centro de Excelencia en Nuevos Materiales (CENM), Universidad del Valle, Cali 760 032, Colombia; 3Instituto Interdisciplinario de las Ciencias, Universidad del Quindío, Armenia 630 004, Colombia

**Keywords:** silver iodide, molecular dynamics, pairwise distribution function, velocity autocorrelation function, (NVE) statistical ensemble, effective charge

## Abstract

The superionic conductor, solid state, and body-centered cubic structure, silver iodide at room temperature, has been studied via molecular dynamics simulations. The calculated results using pairwise Coulomb-Buckingham potential, zero pressure on the sample, a semi-rigid model system of 1000 Ag and 1000 I ions, (NVE) as a statistical ensemble, and an effective charge of Z=0.63 for the pairs Ag-Ag and I-I, were found to be consistent with experimental data and one study using Z=0.60, different potential, and simulation software. For the pair Ag-I, there is a discrepancy due to the high silver ion diffusion. The calculated value of the diffusion constant of the silver ion is greater than iodide ion. The dynamic transport properties (mean square displacement, velocity autocorrelation function) results indicated typical behavior reported by other authors, using different potentials in their DM simulations for iodine and silver ions.

## 1. Introduction

Silver iodide (AgI) is a superionic conductor with high ionic conductivity for temperatures above transition temperature $T_{t}=420$ K [1]. From a technological point of view, this system has good potential for technological applications in fuel cells, lightweight lithium-ion batteries, electrochemical power devices, supercapacitors, among others [2,3,4,5,6]. From a theoretical point of view to understanding the transport dynamics in ionic conductors.

When the temperature is increased from room temperature to $T_{t}$, AgI has the metastable γ-phase (face-centered cubic structure of the zinc blend type) and the β-phase (hexagonal structure of the wurtzite type). At $T_{t}$, AgI undergoes a first-order phase transition from β-phase to α-phase (body-centered cubic structure) with a unit cell containing one silver ion and one iodide ion [7]. Above 420 K, α-AgI is a highly disordered phase in the cationic distribution and a rigid structure provided by the anion sublattice [8]. The mobile silver ions move throughout 42 available sites, leading to high ionic conductivity values.

Theoretically, Sato and Kikuchi used the path probability method to unify three different kinds of interactions between ordering and ionic conductivity [9]; Sunandana and Kumar described that cluster formation and strong interaction between moving ions and clusters explain the first-order phase transition to the α-AgI or high-temperature phase [10]; He et al., showed that fast diffusion does not occur through isolated ion hopping but “through concerted migrations of multiple ions with low energy barrier” [11]. Many authors have reported studies of simulations such as the methods of Monte Carlo and molecular dynamics (MD) [12,13,14,15,16,17,18,19,20,21] to understand the behavior of ionic transport dynamics from a microscopic perspective and by considering some potential interactions between the ions of the AgI system.

The MD method is deterministic, based on solving Newton’s equations of motion for a molecular system. This computer simulation technique explores macroscopic properties by providing information at the microscopic level of a system, i.e., it studies how molecules move, deform, and interact during a certain (computational) time. Depending on the system, several potentials have been proposed by various methods, such as ab initio and classical phenomenological methods, among others [22,23]. Thermodynamic properties depend on the statistical ensemble; properties characterizing the local structure of the system (static structural) are obtained from the pair distribution function (PDF), g(r), and mean square displacement (MSD). Dynamic properties are derived from temporal correlation functions such as the velocity autocorrelation function (VAF).

The first MD simulations in AgI begin with Schommers with 256 particles, a Coulomb–Born–Mayer-type potential, and a temperature of 563 K, showing that correlations between different ions must be distinctly reflected in frequency-dependent conductivity [12]. Using Pauling’s ideas about ionic radii, Vashishta and Rahman (VR) constructed an ion potential model of two terms: Coulomb plus charge-dipole interactions, with 256 particles, reported the diffusion constant of the silver ion (Ag${}^{+}$), the PDF or g(r), and the Ag${}^{+}$ density map [13]. With 500 particles and a soft core potential, Endo and Endo studied VAF [14]. Fukumoto et al., with Z=|0.6|, a potential Lennard-Jones, LJ, and at temperatures of 700, 1000, 1300, and 1500 K, characterized the ionic movements, VAF, and g(r) [15]. Parrinello et al., studied the phase transition α⇌β using a new MD technique that allowed variation in the shape and size of the cell [16]. Tallon simulated the phase diagram using the constant stress MD simulations based on the Parrinello-Rahman Lagrangean [17]. Lee et al. studied the β→α phase transition with the (NVE) ensemble using a VR-type potential obtaining the g(r) at various temperatures, but the transition was not found. With the (NVE), some evidence of phase transition was found [18]. Bitrián and Trullàs simulated the solid phase α-AgI, at T=573 K, using the rigid ion model (with a VR type potential) and two polarizable ions models; superionic conductivity presented better agreement with experimental data for α-AgI and molten AgI for the two polarizable models [19]. Ahmad et al., used optimized LJ potential to reproduce the zinc blend structure of AgI [20]. Vivas et al., studied the microscopic dynamics using Buckingham potential and a (NVE) ensemble for the increasing *T* and (NPT) for the decreasing *T* [21].

The current work compares static structural properties [g(r) and its integration or coordination number $N_{ij}\left(r\right)$] and dynamics (MSD, VAF) obtained by DLPOLY4 for three different values of effective charge at room temperature of the AgI superionic system with the results reported by other authors. Using MSD, the diffusion constant is calculated. Choosing a suitable value for the effective charge leads to obtaining values of the distances of the representative peaks of the different pairs (Ag-Ag, Ag-I, and I-I), which agree with the experimental data.

## 2. Results and Discussion

The CB pair potential is given by Equation (Equation 1):(1)$$V_{ij}\left(r\right)=\frac{k_{C}\;Z_{i}Z_{j}\;e^{2}}{r}+A_{ij}\exp\left(-\frac{r}{\rho_{ij}}\right)-\frac{C_{ij}}{r^{6}}$$
where the first term defines the long-range electrostatic interaction between two ions (with effective charges $Z_{i}$ and $Z_{j}$), $k_{c}={\left(4\pi \varepsilon_{0}\right)}^{-1}$ is the Coulomb constant (with $\varepsilon_{0}$ the vacuum permittivity), and the last two terms represent the non-Coulombic contribution between each pair of ions. $A_{ij},\rho_{ij}$, and $C_{ij}$ are adjustment parameters that depend on the system to study. The Ewald sum technique [24] was used to compute the long-range nature (electrostatic interactions). The values of the adjustment parameters for each pair of ions (represented by the ij subscripts) are listed in Table 1 [25].

For the first term of the potential (Equation 1), the Coulomb interaction must be cut to a finite distance, called cut-off distance $r_{c}$, between ions to reduce the number of iterations in each time step. Typically, the value of $r_{c}$ is about 2.5σ to 5.0σ (σ is the ionic radius). The cut-off distances for two short-range potential terms (Equation 1) (electrostatic repulsion and the London dispersion contribution) and Coulombic interactions were $r_{c}=10$ Å. Figure 1a illustrates the shape of the potential for the different interactions between the ions Ag and I: the continuous black line is for the pair Ag-I, red is for Ag-Ag, and blue is for I-I. For both Z=|1.00| and Z=|0.63|, the potential (Equation 1) falls to zero at $r_{c}=4.5$ Å, as shown in Figure 1a,b, respectively.

The distance estimation between centers of two randomly distributed ions or the measurement of their proximity is called PDF, and the calculation of this function serves to obtain the structure of the system under study. Figure 2a shows two slightly separated peaks, corresponding to the pair Ag-Ag in $r_{\mathrm{Ag}-\mathrm{Ag}}=2.37$ Å (black line) and $r_{\mathrm{Ag}-\mathrm{Ag}}=2.42$ Å (red line) for non-traditional effective charge values $Z_{\mathrm{Ag},\mathrm{Ag}}=\left|0.60\right|$ and $Z_{\mathrm{Ag},\mathrm{Ag}}=\left|0.63\right|$, respectively. For a traditional case, the peak is at $r_{\mathrm{Ag}-\mathrm{Ag}}=4.67$ Å (blue line).

For $r_{\mathrm{Ag}-\mathrm{Ag}}<2.07$ Å, according to the strong repulsive forces, g(r)=0. The first peak is sharp and large, and the last few peaks positioned in regular intervals are broad and oscillate around one until gradually approaching one, indicating no long-range order (see dotted line in Figure 2a), characterizing a typical solid [26,27]. This behavior is consistent for $Z_{\mathrm{Ag},\mathrm{Ag}}=\left|0.60\right|$ and $Z_{\mathrm{Ag},\mathrm{Ag}}=\left|0.63\right|$. However, for $Z_{\mathrm{Ag},\mathrm{Ag}}=\left|1.00\right|$, the PDF does not tend to be one. The coordination number $N_{ij}\left(r\right)$ (proportional to the PDF integral) is the area under the corresponding curve; therefore, for $Z_{\mathrm{Ag},\mathrm{Ag}}=\left|0.60\right|$, $N_{\mathrm{Ag}-\mathrm{Ag}}=7$; $Z_{\mathrm{Ag},\mathrm{Ag}}=\left|0.63\right|$, $N_{\mathrm{Ag}-\mathrm{Ag}}=7$ and $Z_{\mathrm{Ag},\mathrm{Ag}}=\left|1.00\right|$, $N_{\mathrm{Ag}-\mathrm{Ag}}=7$. Figure 2b shows the respective results.

Table 2 summarizes the values obtained by different authors for the distance between an Ag ion and its first neighbors (Ag ions) and their respective effective charge values. In the first two rows, the authors used different potentials (VR and Buckingham), and the last rows are for this work and experimental data.

Figure 3a shows the PDF behavior for the pair Ag-I (I-Ion is assumed to be the reference, and its neighbors are Ag-Ions); the first peak is at $r_{\mathrm{Ag}-I}=2.87$ Å for $Z_{\mathrm{Ag},I}=\left|1.00\right|$ (blue line) and both $Z_{\mathrm{Ag},I}=\left|0.63\right|$ (red line) or $Z_{\mathrm{Ag},I}=\left|0.60\right|$ (black line) peaks are superimposed at $r_{\mathrm{Ag}-I}=3.12$ Å. The curves $Z_{\mathrm{Ag},I}=\left|0.60\right|$ and |0.63| show behavior more adjusted to the experimental data. The blue line shows similar behavior to the Ag-Ag PDF, i.e., there are correlations between Ag and I ions due to the interionic interactions. Ions with values Z=1.00,0.63, and 0.60 occupy the same volume to calculate g(r) [28]. The coincidence of the black and red lines is due to the slight difference in *Z*, of about 5% between 0.63 and 0.60. Figure 3b shows the coordination number $N_{\mathrm{Ag}-I}=8$ for $Z_{\mathrm{Ag},I}=\left|0.60\right|$; $N_{\mathrm{Ag}-I}=8$ for $Z_{\mathrm{Ag},I}=\left|0.63\right|$ and $N_{\mathrm{Ag}-I}=9$ for $Z_{\mathrm{Ag},I}=\left|1.00\right|$.

Table 3 illustrates the comparison with other work reported for the Ag-I pair.

Figure 4a shows the PDF behavior for the pair I-I as a function of *r*; an overhanging peak is presented at $r_{\mathrm{II}-I}=4.67$ Å for $Z_{I,I}=\left|1.00\right|$ and two very close peaks at $r_{\mathrm{II}-I}=4.37$ Å for $Z_{I,I}=\left|0.60\right|$ and $r_{\mathrm{II}-I}=4.42$ Å for $Z_{I,I}=\left|0.63\right|$. This behavior shows a possible diffusion of the iodide ion; however, the blue line has the same behavior of Ag-Ag PDF, indicating a correlation between I-I ions. The coordination number for $Z_{I,I}=\left|0.60\right|$ is $N_{I,I}=7$; $Z_{I,I}=\left|0.63\right|$, $N_{I,I}=7$ and $Z_{I,I}=\left|1.00\right|$, $N_{I,I}=7$, is displayed in Figure 4b.

Table 4 illustrates the comparison with other work reported for the Ag-I pair.

Comparing Table 2, Table 3 and Table 4, our results for the pairs Ag-Ag and I-I are according to the experimental data [2], and are similar to Ref. [16]. For the pair Ag-I, there is a discrepancy due to the high silver ion diffusion.

The rate at which charges move in AgI is described by the time-dependent correlation function MSD of the silver and iodine ions. The MSD determines the average distance of silver and iodide ions, which can be computed at time *t* following Equation (Equation 2):(2)$$\mathrm{MSD}\equiv \frac{1}{N}\sum_{i=1}^{N}{\left|r_{i}\left(t\right)-r_{i}\left(0\right)\right|}^{2}$$
where *N* is the number of particles to be averaged, *r* their coordinates with $r_{i}\left(0\right)$ as the reference position, and $r_{i}\left(t\right)$ the position at time *t*.

Figure 5 displays the behavior of the MSD as a function of time (in ps) for iodide (black line) and silver (red line) ions for the AgI system at 300 K, 0.63 as effective charge, 2000 particles or 1000 Ag${}^{+}$, and 1000 I${}^{-}$. At t=10 ps, the displacements for silver and iodide ions are 1.24 Å and 0.78 Å, respectively, showing larger displacement for Ag${}^{+}$ compared to I${}^{-}$. The behavior of the Ag ion is expected to be highly linear as computational time increases in contrast with the iodide ion’s linear behavior (with a slight slope).

The ballistic regime contribution to the MSD is approximately $t^{2}$ and corresponds to the first four and two picoseconds for Ag${}^{+}$ and I${}^{-}$, respectively. After t=2 ps, the iodide ion has linear behavior with a slight slope, while the silver iodide, after t=4 ps, is more significant.

From the MSD results for the AgI system (three-dimensional and assuming isotropy), one can calculate the diffusion coefficient, defined in Equation (Equation 3):(3)$$D_{\mathrm{Ag}}=\frac{1}{6}\lim_{t\to \infty }\frac{d}{dt}\mathrm{Ag}-\mathrm{MSD}$$

In the first approximation, the slope of Figure 5 is proportional to $D_{\mathrm{Ag},I}$ and depends on the MD time step used or the system. The value for the Ag ion is $2.14\times 10^{-4}$ cm ${}^{2}$/s, which is the same order as the experimental value reported at 300 K [29]. For the I ion, it is $0.95\times 10^{-4}$ cm${}^{2}$/s. The Ag ions are more mobile than I ions, similar to but more relevant than the behavior shown in the work of Vivas et al. [21], who used Z=0.6, T=390,420,460,760 K, 1728 Ag ions, and 1728 I ions. The MD step was in femtoseconds.

Another time-dependent correlation function is the VAF which reveals a variety of dynamical processes originate at a microscopic scale in a molecular system, connecting with measurable macroscopic quantity as the diffusion coefficient.

Figure 6 exhibits typical behavior of the VAF as a function of time (in ps) for iodide (black line) and silver (red line) ions. Both curves have characteristic behavior: near t=0 ps, the velocity presents the highest correlation; at t=0.24 ps, iodide ions reach their minimum value, whereas silver ions are at t=0.31 ps. The black curve for t=0.43 ps and the red curve for t=0.75 ps tends to be zero and oscillate asymptotically. Negative VAF values mean that the ions interact strongly with their neighbors, and this interaction keeps them vibrating in a bounded region of space.

The negative VAF values for iodine ions are higher for silver ions, evidencing a higher stiffness tendency. In other words, the silver ions are in the solid state with a transport dynamic “liquid-like” over a “rigid lattice” formed by iodine ions. Figure 2, Figure 3 and Figure 4 display this behavior, where the g(r) are almost similar, showing a solid state (by the defined peaks) and oscillation around one, characteristic of a liquid.

## 3. Materials and Methods

Simulations were performed with the parallel MD simulation package DL_POLY_4 [30]. Assuming the crystal structure of AgI as a body cubic centered (bcc) lattice, the constructed simulation cell containing 2000 ions (Ag${}^{+}$ = 1000, I${}^{-}$ = 1000) in 10×10×10, lattice parameter a=46.9584, and cubic periodic boundary conditions was applied. We chose (NVE) as the statistical ensemble, since the characteristics of the system are: not performing any mechanical work and not exchanging even heat; it is a simple physical phenomenon (calculate g(r),N(r), MSD, and VAF for a particular value effective charge). The discrete-time steps in MD integrals are of the order of 1–10 ps (there are no long MD simulations) [31], zero pressure on the sample, and the isotherm was kept at room temperature (300 K). The MD steps are $1.5\times 10^{5}$, the first $0.1\times 10^{5}$ is used to equilibrate the system, and 0.001 ps is the MD time step.

Crystal structure data were obtained from [32], and the evidence of the atomic positions from [33]. For clarity, the AgI system is a solid electrolyte, but its transport dynamics is such that the motion of silver ions can be considered “liquid-like” in the lattice of anions [34].

In the study of MD simulations, information on the characteristics of ionic systems depends mainly on a “good” choice of potential for interaction between ion pairs of the system to be studied. In our case, the considered interactions are those of the ion pairs I-I, Ag-Ag, and Ag-I. Deformation of the electron structure of the ions is generally not taken into account, i.e., the ions are considered semi-solid spheres interacting through an effective charge [35] in the pair potential [36]. The effective pair potential chosen for the current work is the Coulomb-Buckingham (CB)-type potential, and we are assuming a semi-rigid ion model [37,38] with traditional (Z=|1.00|) and non-traditional (Z=|0.60| and Z=|0.63|) effective charges.

## 4. Conclusions

In this work, we performed molecular dynamics simulations using the DL_POLY_4 package with the potential of Coulomb-Buckingham, obtaining both static and dynamic properties for silver iodide for Z=|0.63|. We report distances Ag-Ag of 2.42 Å and I-I of 4.42 Å, in agreement with the experimental data and one study using Z=0.60, different potential, and simulation software. The distance for Ag-I, of 3.12 Å, contained discrepancies in terms of the high iodide ion diffusion. We found that the coordination number for the pair Ag-I depends slightly on the choice of the effective charge value, while for the pairs Ag-Ag and I-I, it is the same.

The MSD displayed that the displacement for the Ag ions is, considering the taken temperature, more significant than iodide ions, showing typical behavior of a solid for AgI. The slope of the MSD curve for Ag ions is greater than that of iodide ions; therefore, the Ag ions have a diffusion coefficient equal to $2.14\times 10^{-4}$ cm${}^{2}$/s, comparable to the experimental value reported at 300 K.

The VAF showed typical behavior reported by other authors, using different potentials in their DM simulations for iodine and silver ions; however, the minimum for silver is higher than those reported. The ionic conductivity was not calculated because no simulations were performed for other temperature values.

## Figures and Tables

**Figure 1 molecules-27-06132-f001:**
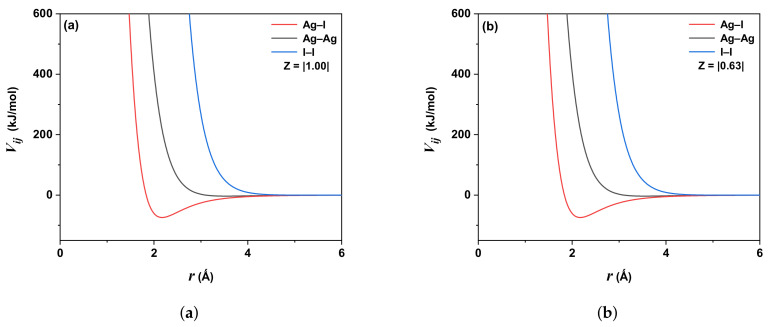
Behavior of the CB-type potential (Equation 1) for the three different interactions: Ag-I (solid red line), Ag-Ag (solid black line), and I-I (solid blue line). (**a**) for Z=|1.00|. (**b**) For Z=|0.63|.

**Figure 2 molecules-27-06132-f002:**
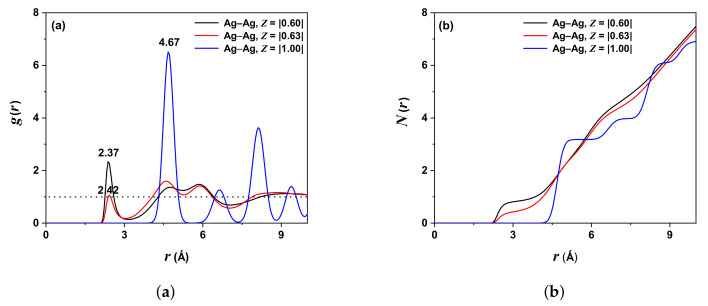
(**a**) PDF or $g_{\mathrm{Ag}-\mathrm{Ag}}\left(r\right)$ as a function of the distance $r_{\mathrm{Ag}-\mathrm{Ag}}$, with $Z_{\mathrm{Ag},\mathrm{Ag}}=\left|0.60\right|$ (black line), $Z_{\mathrm{Ag},\mathrm{Ag}}=\left|0.63\right|$ (red line) and $Z_{\mathrm{Ag},\mathrm{Ag}}=\left|1.00\right|$ (blue line). (**b**) Coordination number $N_{\mathrm{Ag}}$ obtained by integrating PDF, at T=300 K.

**Figure 3 molecules-27-06132-f003:**
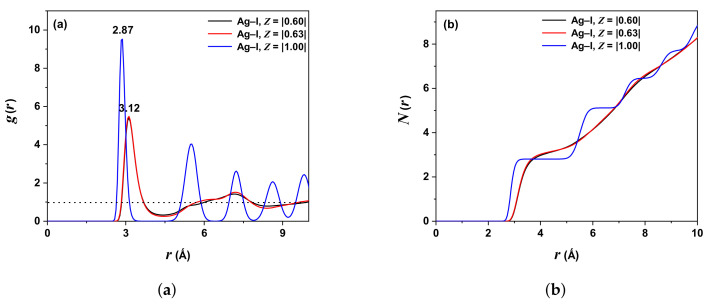
(**a**) PDF (Ag-I) with $Z_{\mathrm{Ag},I}=\left|0.60\right|$ (black line), $Z_{\mathrm{Ag},I}=\left|0.63\right|$ (red line), and $Z_{\mathrm{Ag},I}=\left|1.00\right|$ (blue line). (**b**) Integrated PDF ($N_{\mathrm{Ag}-I}$). Temperature 300 K.

**Figure 4 molecules-27-06132-f004:**
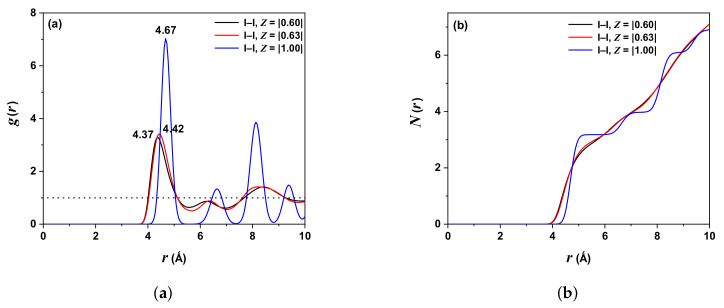
(**a**) PDF (I-I) with $Z_{I,I}=\left|0.60\right|$ (black line), $Z_{I.I}=\left|0.63\right|$ (red line), and $Z_{I,I}=\left|1.00\right|$ (blue line). (**b**) Integrated PDF ($N_{I-I}$). Temperature 300 K.

**Figure 5 molecules-27-06132-f005:**
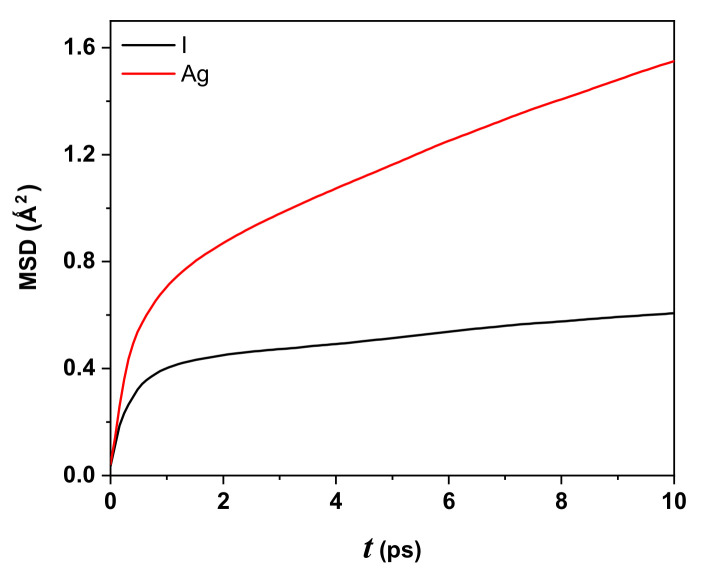
The behavior of the MSD of AgI-system for silver ion (red line) and iodide ion (black line) with respect to time at T=300 K.

**Figure 6 molecules-27-06132-f006:**
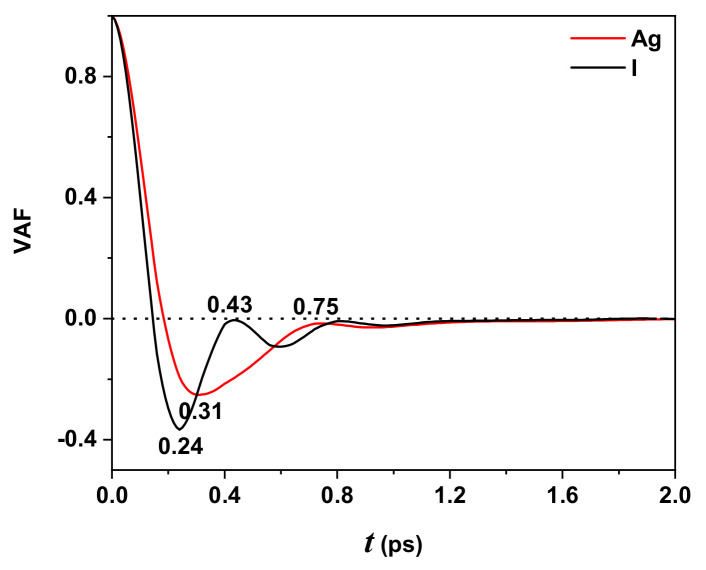
VAF for Ag (red line) and I (black line) ions. The black line has a minimum at t=0.24 ps, while the red line at t=0.31 ps. After t=0.5 ps, both curves oscillate around zero. Temperature 300 K.

**Table 1 molecules-27-06132-t001:** Values of adjustment parameters for pair potential (Equation 1).

	$Z_{\mathrm{ij}}$	$A_{\mathrm{ij}}$ (kJ/mol)	$\rho_{\mathrm{ij}}$ (Å)	$C_{\mathrm{ij}}$ (kJ·Å${}^{6}$/mol)
I-I	−0.63/−0.63	4,531,732.7	0.3097	9648.5
Ag-Ag	0.63/0.63	1,594,709.6	0.2310	21,612.7
Ag-I	0.63/−0.63	512,510.8	0.2979	13,758.8

**Table 2 molecules-27-06132-t002:** Comparison of Ag-Ag PDF. The temperature at 300 K.

Ag-Ag	$r_{\mathrm{ij}}$ (Å)	$Z_{\mathrm{ij}}$	Ref
	2.80	0.6	[16]
	4.20	1.0	[21]
	2.37/2.42	0.60/0.63	This work
	2.40		[1]

**Table 3 molecules-27-06132-t003:** Comparison of Ag-I PDF. The temperature at 300 K.

Ag-I	$r_{\mathrm{ij}}$ (Å)	$Z_{\mathrm{ij}}$	Ref
	2.80	0.6	[16]
	3.10	1.0	[21]
	3.12	0.60/0.63	This work
	2.8		[1]

**Table 4 molecules-27-06132-t004:** Comparison of I-I PDF. The temperature at 300 K.

I-I	$r_{\mathrm{ij}}$ (Å)	$Z_{\mathrm{ij}}$	Ref
	4.58	0.6	[16]
	4.2	1.0	[21]
	4.37/4.42	0.60/0.63	This work
	4.4		[1]

## Data Availability

Not applicable here.

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
