# Peer review of "Molecular Dynamics Study of Structural and Transport Properties of Silver Iodide Using Effective Charges"

_molecules, 2022, doi:10.3390/molecules27186132_

Round 1
Reviewer 1 Report
The manuscript of Lara et al. describes molecular dynamics studies of crystalline AgI. It seems that, via using effective ionic charges, these authors are able to reproduce experimental results on the partial PDF -- I therefore am not against an eventual publication. Before that, however, a number of issues must be addressed.
(1) Both the reason why this study was need and the main conclusion should be stated more clearly.
(2) My main concern is that, just by looking at he partial PDF-s, one cannot be sure that the simulated system is still crystalline: I think that at least the I-I pPDF should show clear signs of crystallinity (long range ordering, for instance). The authors must provide proof, e.g., a plot of the atomic positions, or the structure factor (structural information in the reciprocal space), etc...
(3) I do not understand why the (NVE) ensemble was chosen: a well known drawback, for instance, the temperature drift that practically almost always occurs. The authors should explain their choice of ensemble.
(4) Still concerning the simulation itself, the time step of 0.01 ps (10 fs !) sounds really too long.
(4) In Figure 1b, one has the feeling that the assignation (colour codes) of the curves is incorrect. If not then some discussion is needed as to why the I-I potential function has swapped with the Ag-I one.
(5) What charges have been used for calculating the VAFs (Figure 5)?
Author Response
Dear
Ms. Miranda Yan
Editor, MDPI AG
Here are the answers to the referee's report. We thank the referee for their comments.
Please find enclosed the revised version of our manuscript, modified by the comments. All the points have their respective answers. Our methodology is the following: paragraph in yellow corresponds to the comments, in green our answers, and in blue the corrected text.

Reviewer 2 Report
Comments from Reviewer
Title: Molecular Dynamics study of structural and transport properties of silver iodide using effective charges.
The current form's presentation of methods and scientific results is satisfactory for publication in the Molecules journal. The minor and significant drawbacks to be addressed can be specified as follows:
1. Abstract. "The total number of temporal (…)". This sentence should be removed. Too much detail about the calculations.
2. Fig. 1. (i) Panel (a) Repeated "(a)". (ii) "Ag" is bolded in contrary to "I". Why?
3. Fig. 1, figure captions. For Z = |0.60| or |0.63| ---> For Z = |0.63|.
4. Fig. 1. A strange sequence of curves- panel (a) Ag-Ag, Ag-I, and I-I, but panel (b) Ag-Ag, I-I, and Ag-I. Why?
5. Why do the authors analyze such close values of Z, i.e. |0.60| or |0.63|? Why not, for example, |0.53|?
6. Fig. 3, figure captions. Please add the information that the red and black lines coincide.
7. Tabs. 3 and 4. The obtained results are confusing. I-I and Ag-Ag: high agreement compared to [1] as opposed to [16] and [21]. Ag-I: low agreement compared to [1] as opposed to [16]!!!
Sincerely,
The reviewer.
Author Response

(The authors gave the same response as above.)

Round 2
Reviewer 1 Report
This is my second reading of this manuscript. It has improved, but I still have my major point unsettled: the authors still need to demonstrate that the system they simulate is not a liquid. The authors' reply was not convincing enough, Figure 4 still suggests that even the I-I prdf is that of a liquid (and, to be honest, Ref. 20 is also not really convincing for me).
For this, I request them to calculate the diffusion constant for both Ag and I ions. If they are (or at least one of them) is sufficiently low then I would make no further objections to publication.
Also, a picture showing the I ions in the simulation box would be helpful
Author Response
According to your right comments, we have calculated the diffusion constant for Ag and I ions using the mean square displacement function.
In Abstract.
The calculated value of the diffusion constant of the silver ion is greater than iodide ion. The dynamic transport properties (mean-square displacement, velocity
(pag. 1, lines 7−9.)
In Introduction section
dynamics (MSD, VAF).
Using MSD, the diffusion constant is calculated.
(pag. 2, lines 71−74.)
Results and Discussion
The rate at which charges move in AgI is described by the time-dependent correlation function MSD of the silver and iodine ions. The MSD determines the average distance of silver and iodide ions followed, which can be computed at time t as:
where N is the number of particles to be averaged, r their coordinates with ri(0) as the reference position, and ri(t) the position at time t.
Figure 5 displays the behavior of the MSD as a function of time (in ps) for iodide (black line) and silver (red line) ions for the AgI-system at 300 K, 0.63 as effective charge, 2 000 particles or 1 000 Ag+ and 1 000 I−. At t = 10 ps, the displacements for silver and iodide ions are 1.24 A and 0.78 A, respectively, showing a larger displacement for Ag+ compared to I−. The behavior of the Ag ion is expected to be highly linear as computational time increases in contrast with the iodide ion’s linear behavior (with a slight slope).
The ballistic regime contribution to the MSD is approximately t2 and corresponds to the first four and two picoseconds for Ag+ and I−, respectively. After t = 2 ps, the iodide ion has a linear behavior with a slight slope, while the silver iodide, after t = 4 ps, is more significant.
From the MSD results for the AgI system (three-dimensional and assuming isotropy), one can calculate the diffusion coefficient, defined as:
(pag. 6, lines 160−173.)
In the first approximation, the slope of figure 5 is proportional to DAg,I and depends on the MD time step used or the system. The value for the Ag ion is 2.14 × 10−4 cm2/s, which is the same order as the experimental value reported at 300 K [38]. For the I ion is 0.95×10−4 cm2/s. The Ag ions are more mobile than I ions, similar to but more relevant behavior of Vivas et al. [21] work, who used Z = 0.6, T = 390, 420, 460, 760 K, 1 728 Ag ions, 1 728 I ions, and the MD step was femtoseconds.
Another time-dependent correlation function is the VAF which reveals a variety of dynamical processes originate at a microscopic scale in a molecular system, connecting with measurable macroscopic quantity as the diffusion coefficient. Figure 6 exhibits typical behavior of the VAF as a function of time (in ps) for iodide (black line) and silver (red line) ions. Both curves have a characteristic behavior: near t = 0 ps, the velocity presents the highest correlation; at t = 0.24 ps, iodide ions reach their minimum value, whereas silver ions are at t = 0.31 ps. The black curve for t = 0.43 ps and the red curve for t = 0.75 ps tends to be zero and oscillate asymptotically. Negative VAF values mean that the ions interact strongly with their neighbors, and this interaction keeps them vibrating in a bounded region of space.
The negative VAF values for iodine ions are higher for silver ions, evidencing a higher stiffness tendency. In other words, the silver ions are in the solid state with a transport dynamic “liquid-like" over a “rigid lattice" formed by iodine ions. Figs. 2, 3, and 4 display this behavior, where the g(r) are almost similar, showing a solid state (by the defined peaks) and oscillation around 1, characteristic of a liquid.
(pag. 7, lines 174−195.)
Conclusions section
The MSD displayed that the displacement for the Ag ions is, considering the taken 206 temperature, very most significant than iodide ions, showing a typical behavior of a solid for AgI. The slope of the MSD curve for Ag ions is greater than iodide ions; therefore, the Ag ions have a diffusion coefficient equal to 2.14 × 10−4 cm2/s, comparable to the experimental value reported at 300 K.
(pag. 8, lines 205−209.)
Some comments
Crystal structure data was obtained from [26], and the evidence of the atomic positions from [27]. For clarity, the AgI system is a solid electrolyte, but its transport dynamics is such that the motion of silver ions can be considered “liquid-like" in the lattice of anions [28].
(pag. 3, lines 89−92.)
For rAg–Ag < 2.07 A, by the strong repulsive forces, g(r) = 0. The first peak is sharp and large, and the last picks positioned in regular intervals are broad and oscillate around 1 until gradually approaching 1, indicating no long-range order. [see dotted line in Figure 2(a)], characterizing a typical solid [35,36].
(pag. 4, lines 126−129.)
The blue line shows similar behavior to the Ag–Ag PDF, i.e., there are correlations between Ag and I ions due to the interionic interactions.
(pag. 5, lines 143−145.)
This behavior shows a possible diffusion of the iodide ion, however, the blue line has the same behavior of Ag-Ag PDF, indicating a correlation between I–I ions
(pag. 5, lines 153−155.)
Additional references
- Briant, L. D.; H. A. Leason. High-Pressure Polymorphs in the Silver Iodide Phase Diagram. Science 1964 146, 519–521.
- Funke, K. Dynamics of Mobile Ions in Materials with Disordered Structures –the Case of Silver Iodide and the Two Universalities. In: Kreysa, G., Ota, Ki., Savinell, R.F. (eds) Encyclopedia of Applied Electrochemistry. Springer, New York, 2014.
- Kvist, A.; Tarneberg R. Self-diffusion of Silver Ions in the Cubic High Temperature Modification of Silver Iodide. Z. Natur. 1970,

Reviewer 2 Report
My comments have been appropriately addressed in the revised manuscript.
Author Response
The referee's comments have been addressed in our revised manuscript.